# Improving the reliability of cohesion policy databases

**Samuele Lo Piano**[1]*, **Emanuele Borgonovo**[2], **Arnald Puy**[3,4], **Andrea Saltelli**[5], **John Walsh**[6], **Daniele Vidoni**[7]

**1** School of the Built Environment, University of Reading, Reading, Berkshire, United Kingdom, **2** Department of Decision Sciences and BIDSA, Bocconi University, Milano, Italy, **3** Department of Ecology and Evolutionary Biology, Princeton University, Princeton, New Jersey, United States of America, **4** Centre for the Study of the Sciences and the Humanities (SVT), University of Bergen, Bergen, Norway, **5** Barcelona School of Management, Universitat Pompeu Fabra, Barcelona, Catalonia, Spain, **6** Directorate-General for Regional and Urban Policy, European Commission, Brussels, Belgium, **7** Directorate-General for Competition, European Commission, Brussels, Belgium

* s.lopiano@reading.ac.uk

## Abstract

In this contribution, we present an innovative data-driven model to reconstruct a reliable temporal pattern for time-lagged statistical monetary figures. Our research cuts across several domains regarding the production of robust economic inferences and the bridging of top-down aggregated information from central databases with disaggregated information obtained from local sources or national statistical offices. Our test bed case study is the European Regional Development Fund (ERDF). The application we discuss deals with the reported time lag between the local expenditures of ERDF by beneficiaries in Italian regions and the corresponding payments reported in the European Commission database. Our model reconstructs the timing of these local expenditures by back-dating the observed European Commission reimbursements. The inferred estimates are then validated against the expenditures reported from the Italian National Managing Authorities (NMAs) in terms of cumulative monetary difference. The lower cumulative yearly distance of our modelled expenditures compared to the official European Commission payments confirms the robustness of our model. Using sensitivity analysis, we also analyse the relative importance of the modelling parameters on the cumulative distance between the modelled and reported expenditures. The parameters with the greatest influence on the uncertainty of this distance are the following: first, how the non-clearly regionalised expenditures are attributed to individual regions; and second, the number of backward years that the residuals of the yearly payments are spread onto. In general, the distance between the modelled and reported expenditures can be further reduced by fixing these parameters. However, the gain is only marginal for some regions. The present study paves the way for modelling exercises that are aimed at more reliable estimates of the expenditures on the ground by the ultimate beneficiaries of European funds. Additionally, the output databases can contribute to enhancing the reliability of econometric studies on the effectiveness of European Union (EU) funds.

**Data Availability Statement:** The data used in the paper can be produced through the code made available from a public GitHub repository: https://github.com/Confareneoclassico/Expenditure_modelling. The data against which we have

validated our findings are publicly available from the platform 'OpenCoesione': https://opencoesione. gov.it/en/. A copy is also available from the GitHub repository as a spreadsheet under the name "Database_Final_UPD_2020_corr.xlsx".

**Funding:** AP worked on this paper on a Marie Sklodowska-Curie Global Fellowship (Horizon 2020 https://ec.europa.eu/programmes/horizon2020/), grant number 792178. The funders had no role in study design, data collection and analysis, decision to publish, or preparation of the manuscript.

**Competing interests:** The authors have declared that no competing interests exist.

**Abbreviations:** **Apl**, Apulia; **AsV**, Aosta Valley; **CF**, Cohesion Fund; **Clb**, Calabria; **Cmp**, Campania; **DG REGIO**, Directorate-General for Regional and Urban Policy; **EAFRD**, European Agricultural Fund for Rural Development; **EC**, European Commission; **EmR**, Emilia-Romagna; **ERDF**, European Regional Development Fund; **ESF**, European Social Fund; **ESIF**, European Structural and Investment Funds; **EU**, European Union; **FVG**, Friuli Venezia Giulia; **IT**, Italy; **Laz**, Lazio; **Lgr**, Liguria; **Lmb**, Lombardy; **Mls**, Molise; **MS**, Member State; **NMAs**, National Managing Authorities; **NUTS**, Nomenclature of Territorial Units for Statistics; **OPs**, Operational Programmes; **QoI**, Quantity of Interest; **Scl**, Sicily; **Srd**, Sardinia; **Tsc**, Tuscany; **TST**, Trentino-South Tyrol.

## Introduction

In this contribution we propose a data-driven model to estimate the actual economic time series from those reported in official centralised databases and benchmark them against bottom-up evidence coming from local statistical offices. The test bed case study is the ERDF, which aims to strengthen economic, social, and territorial cohesion in the EU by correcting interregional imbalances. ERDF is part of the European Structural and Investment Funds (ESIF), which represent roughly two-thirds of the whole European budget, amounting to more than 1 trillion euros for the period 2014–2020 (approximately 1% of the EU-28 gross national income). For the same peoriod, the ERDF allocated more than 220 billion euros in investments to diverse areas.

Additionally, a new temporary instrument, the NextGenerationEU, amounting to 750 billion euros has been introduced for the period 2021–2027 to ease the recovery of EU countries from the COVID-19 pandemic. The extraordinary circumstances that have arisen due to this challenge further emphasise the importance of understanding the spending pattern of public resources [1], particularly so as to inform the public about the allocations of EU tax-payers' money and their benefits.

Every Member State (MS) in the EU is obliged to record and account for the expenses incurred, but the complete records are neither (always) publicly available nor directly comparable across individual MSs. For this reason, a spatially and temporally homogeneous expenditure database at the European level will always be lacking. This is precisely why resorting to modelling these expenditures can help to alleviate the issue.

The European Commission (EC)-managed database suffers from an inevitable time lag because EC payments are reimbursed only after the incurred expenses have been invoiced. Precisely, the final beneficiaries invest on the ground and produce invoices to the NMAs, which, in turn, certify the share of eligible expenditures and produce an invoice to the EC. Even if this process runs smoothly (i.e., the relevant documents are promptly processed at each stage, no accounting mistakes are made, payments are not suspended following audit inspections, and so on), a substantial time lag may occur from the incurred expenditure on the ground to the moment when the EC payment to the NMAs is actually recorded. Therefore, it is very challenging to produce econometric inferences regarding the benefits of EU cohesion policy funding on the ground.

The model we discuss in this contribution estimates the expenditures incurred by European regions from their observed EC reimbursement pattern. To the best of our knowledge, it is the first time this approach is attempted. Our modelled expenditures are validated against the expenditures reported by the Italian authorities, which have recently become available.

Econometricians can make use of the output database of modelled expenditures to generate more robust inferences regarding the benefits of EU cohesion policy in a number of fields, including convergence analysis [2–6] and in terms of the effects on GDP [7, 8], employment [7], or local administrative and governance capacity [9–11] among many others.

The next section presents the data and methods used for the present study. In addition, we demonstrate how a Wasserstein measure [12], which gives the distance between two curves, is a convenient proxy to drive inferences on the observed reimbursement patterns and to model the target pattern of expenditures.

## Materials and methods

Data for ESIF EC payments to MSs and regional authorities were obtained from the EC Directorate-General for Regional and Urban Policy (DG REGIO). In every five-year financial programming period of the EU, each individual MS systematically records the requests for

payments (and related invoices) made by individual beneficiaries. However, these documents are neither standardised, or necessarily accessible, nor contain the same information or structure across regions or MSs. Furthermore, the number of primary sources amounts to more than 300 because the individual NMAs store their own records, which has made unsuccessful the attempts to build these data bottom up due to the unreliable and incomplete information across MSs.

The current case study is grounded on the "official" database at the EC level. This dataset consists of around 500,000 entries, which include the yearly payments to the Operational Programmes (OPs) over four overarching funding schemes (Cohesion Fund (CF), European Agricultural Fund for Rural Development (EAFRD), ERDF, European Social Fund (ESF)) and five programming periods (1989–1993, 1994–1999, 2000–2006, 2007–2013, 2014–2020). OPs are the reference unit and have variable geographical scope: regional, multi-regional, national, or across multiple MSs. The EC does not directly collect information disaggregated at the regional level.

The pre-processing stage of this study involved regionalising EC payments over the 280 regions in Nomenclature of Territorial Units for Statistics (NUTS), known as NUTS 2. Over time, the number and borders of NUTS2 regions have changed to reflect countries joining or leaving the EU, as well as administrative readjustments within individual MSs. For this reason, during pre-processing, we harmonised the nomenclature to the NUTS2 2010 version, where each code maps uniquely onto a specific region. The detailed regionalisation procedure is described in Wishlade et al. [13] (work-package 13) and Lo Piano et al. [14]. Every entry in this regionalised EC dataset of payments corresponds to an amount reimbursed to a given regional authority over a specific year, funding scheme, and programming period. For the funding scheme ERDF, non-regionalised figures were attributed on a pro-capita basis to individual NUTS2 areas for those countries that had not broken down the information at the NUTS2 level. If the region's level of development was specified, a pro-capita attribution was performed among the NUTS2 areas with the same level of development.

In this contribution, we cover the funding scheme ERDF reimbursed over the programming period 2007–2013. We tested the modelled expenditures against the actual incurred expenditures for a pioneer MS, Italy (IT), whose data were provided by the Italian managing authority for this programming period and funding scheme. These data are available from https://opencoesione.gov.it/en/ and also included uncertified expenditures (i.e., figures that have not yet been accounted for in the consolidated EC payments database).

Sensitivity analysis [15–17] was used to explore how uncertainty in the input variables affected the output variable. Our analysis was performed within a factor-prioritisation [18] and direction-of-change setting; that is, we aimed to identify the key drivers that influenced the output variable and the impact associated with fixing these drivers, respectively. One of the most widely adopted approaches to sensitivity analysis is the variance-based approach, wherein output uncertainty is measured in terms of the statistical moment variance, which is eventually apportioned to the input parameters. Another class of sensitivity methods, the moment-independent sensitivity measures, do not resort to a particular statistical moment. As we will see in the next section, our output Quantity of Interest (QoI) is the distance between the cumulative distributions of the expenditures, which naturally supports the selection of a moment-independent sensitivity measure. Several moment-independent measures have been proposed in the literature (see Borgonovo [19], Plischke, Borgonovo & Smith [20], and Pianosi & Wagener [21]). We hereby adopt the $\delta$ moment-independent sensitivity measure [19, 20] to evaluate how the modelling parameters influence uncertainty regarding the distance between the modelled and the reported expenditures.

## Use of *distance* measures to estimate expenditure patterns

Consider a generic region $p$ whose expenditures can be reimbursed over $k$ eligible years. Let us also introduce a *dummy* region against which we can benchmark the reported reimbursement patterns. The dummy region has a constant spending and reimbursement pattern; specifically, it spends a certain amount and is reimbursed for the same amount each year until the last eligible year of the programming period. For instance, if reimbursements can occur over 10 years, the dummy region is reimbursed 10% each year of the total budget it has been granted.

For the sake of comparability, reimbursements are normalised across regions. The normalised cumulative expenses of the dummy region over $k$ years can be expressed as per Eq 1:

$$D = \sum_{i=1}^{k} d_i = \sum_{i=1}^{k} \frac{1}{k} = 1 \tag{1}$$

Analogously, one can define the regional reimbursement pattern for a generic region $p$ as follows:

$$R_p = \sum_{i=1}^{k} r_{p_i} = 1 \tag{2}$$

One may also define an equivalent of the Wasserstein metric in probability theory as the distance $\mu_p$ between the cumulatives of these two curves [12] according to Eq 3:

$$\mu_p = \sum_{i=1}^{k}\sum_{o=1}^{i} |r_{p_o} - d_o| \tag{3}$$

To assess the time specificity of the reimbursement trend rather than the simple divergence from the regular *dummy* pattern, it is possible to address the plain difference in line with the Kruglov distance [12]. This is denoted by $\mu_p^s$ and can be expressed as follows:

$$\mu_p^s = \sum_{i=1}^{k}\sum_{o=1}^{i} (r_{p_o} - d_o) \tag{4}$$

These measures are complementary: $\mu_p$ acknowledges the divergence from a constant spending pattern, although it does not allow to grasp its time specificity (i.e., early versus late reimbursement pattern); conversely, $\mu_p^s$ addresses this specificity, although it suffers from compensatory effects across years (e.g., a positive difference in the year $i$ would be cancelled out by an equal negative difference in the year $i + 1$). This makes it impossible to evaluate the precise reimbursement pattern at a yearly granularity through $\mu_p^s$.

Let us now analyse the extreme case in which all the EC payments are reimbursed in the last eligible year of the programming period. In this case, the maximum difference $\mu_p$ would be:

$$\mu_p = \sum_{i=1}^{k}\sum_{o=1}^{i} |r_{p_o} - d_o| = \sum_{i=1}^{k-1} |-\frac{i}{k}| = \frac{1}{k}\sum_{i=1}^{k-1} i = \frac{k-1}{2} \tag{5}$$

Analogously, the assessment can be repeated for a hypothetical region whose expenditures are entirely reimbursed in the first year.

$$\mu_p = \sum_{i=1}^{k}\sum_{o=1}^{i}|r_{p_o} - d_o| = \sum_{i=1}^{k}|1 - \frac{i}{k}| = \frac{1}{k}\sum_{i=1}^{k}|k - i| = \frac{1}{k}\sum_{l=0}^{k-1}|l| = \frac{k-1}{2} \qquad (6)$$

The figures shown in Eqs 5 and 6 define the maximum threshold for our measurement. The minimum value zero is attained for a hypothetical region whose reimbursement pattern is identical to the *dummy region*.

One can easily obtain the values for the $\mu_p^s$ measurement, which is equivalent to $\mu_p$ in the case of an early reimbursement pattern. The sign is the opposite in the case of extremely late reimbursement and amounts to $-\frac{k-1}{2}$. Overall, regions anticipating the dummy spending pattern have a positive sign, whereas the sign is negative for regions with a delayed reimbursement pattern.

The two measures are complementary: $\mu_p$ measures the regularity of the reimbursement pattern, which is a property that policymakers will benefit from considering when managing the spending of granted financial resources. $\mu_p^s$ points to the specificity of the regional reimbursement pattern over the course of the entire programming period (delayed vs. early).

We use $\mu_p^s$ to define an index of regional specificity $Ir$ that, in turn, enables us to rank regions and propose regional spending patterns, as detailed in the following section.

## Taxonomy of cases

Let us consider the cumulative annual history of expenditures invoiced from the ultimate beneficiaries to the NMAs, $MS_p$. $E_p$ is our modelled cumulative expenditure, namely the quantity with which we attempt to reproduce the pattern of $MS_p$. As per Eq 7, the sum of the yearly figures over the entire programming period must be equal to assure consistency.

$$\sum_{i=1}^{k}r_{p_i} = \sum_{i=1}^{k-m}ms_{p_i} = \sum_{i=1}^{k-m}e_{p_i} \qquad (7)$$

EC payments are spread over $k$ years, while expenditures are not eligible after the $(k-m)^{th}$ year of the programming period.

A general rule is that reimbursements always follow expenses, which can be used as a basis for modelling the yearly expenditure patterns. A situation wherein cumulative modelled expenditures are smaller than the expenditures of NMAs would be wrong and would require the correction of our model.

For instance, let us take the first year of the programming period. In this year, the relation between the yearly figures must be:

$$r_{p_1} \leq e_{p_1} \qquad (8)$$

Furthermore, $ms_{p_1}$ anticipates $r_{p_1}$, and the relation between these two quantities must also hold.

$$r_{p_1} \leq ms_{p_1} \qquad (9)$$

Finally, significant differences between $ms_{p_1}$ and $e_{p_1}$ would not be plausible. $ms_{p_1}$ also accounts for invoices sent by local authorities (e.g., municipalities). This condition ensures that the time lag with $e_{p_1}$ is minimal and, in particular, below the yearly granularity at which

these figures have been produced. Therefore, the relation between the two figures should be:

$$ms_{p_1} \approx e_{p_1} \tag{10}$$

We can also extend these relations to the $l^{th}$ year, with $l \leq k - m$, as per the following:

$$\sum_{i=1}^{l} r_{p_i} \leq \sum_{i=1}^{l} ms_{p_i} \approx \sum_{i=1}^{l} e_{p_i} \tag{11}$$

Therefore, our workflow will firstly focus on evaluating the closure relation as per Eq 8, although uncertified expenditures may result in discrepancies of variable magnitude.

## Modelling the incurred expenditures

In this contribution, we use an adaptation of the model to estimate local expenditures from the reported EC payments [14, 22, 23] and validate the ERDF figures for Italy against those of the national managing authority. The model was developed based on joint reflections among practitioners in technical fields such as modelling and data analysis, as well as practitioners involved in the operation of EU regional policy programmes. The rationale of the model is to project the reimbursed payments backwards to capture the actual temporality of the reimbursed financial resources, along with their effects on the local receiving areas.

An overarching assumption of the model is that each yearly expenditure corresponds only to a fraction of the payment reimbursed in the same year. The complementary fraction of this payment is attributed to expenditures incurred over the previous year(s).

Yearly expenditures are estimated by ranking each EU region against a dummy region, as discussed in the previous subsections. In turn, a coefficient of regional specificity $Ir_p$ is calculated from the ranks of $\mu^s$ over each individual funding scheme and programming period. This coefficient is used to define the spending pattern of regions. The higher its value, the greater is the delay that characterises the region's reimbursement pattern compared to the dummy region's constant reimbursement pattern.

Feature scaling, also known as min-max normalisation, was performed on the $Ir_p$ series. This leads to a value of 0 for the region with the earliest reimbursement pattern and 1 for the latest.

For instance, consider $Ir_p = 0.6$ for the ERDF funding scheme over the 2007–2013 programming period. This implies that the payment reported in the last eligible year of expenditures is attributed to expenditures that were also incurred over a maximum of $int((2017 - 2007) * Ir_p^{2007-2013,ERDF}) = 6$ previous years.

The uncertain parameters in the model, as shown in Table 1, are:

- Maximum share of payment attributed to an expenditure incurred on the same year $\phi_{max}$

- Minimum share of payment attributed to an expenditure incurred on the same year $\phi_{min}$

**Table 1. Summary of parameters and their distribution.** $\mathcal{D}$ and $\mathcal{U}$ indicate discrete and uniform, respectively.

| Parameter | Description | Distribution |
|---|---|---|
| $\phi_{max}$ | Maximum share | $\mathcal{U}(0.8, 1)$ |
| $\phi_{min}$ | Minimum share | $\mathcal{U}(0.2, 0.4)$ |
| Residual selector | Binary trigger for non-clear NUTS 2 expenditure IT database | $\mathcal{D}(0, 1)$ |
| Years | Number of previous years from the last year of eligible expenditures | $\mathcal{D}(1, integer((k - 1) * Ir_p^{pp,fs}))$ |

- Number of *Years* of expenditures that the residual payment can be attributed to backwards, which is defined as $\text{int}(k-1) * Ir_p^{pp,fs}$ for a generic region $p$ over the programming period $pp$ and the funding scheme $fs$

The other uncertain parameter is a binary trigger related to non-attributed NMAs expenditures. These are expenditures that do not map onto specific NUTS2 areas. This regional reattribution is proportional either to the funds reimbursed on the year of the unattributed payment or to the funds reimbursed over the whole programming period. Table 1 specifies the uncertainty range of these input parameters.

The range of the distributions of the input parameters was selected after consultation with practitioners directly involved in EU regional policy programmes. For continuous variables, uniform distributions were conservatively adopted due to the absence of information concerning the individual probability of values across the range.

The share of the payment attributed to an expenditure occurring in the same year is denoted by $\phi_p$, which is expressed as follows:

$$\phi_p = \phi_{max,p} - Ir_p^{pp,fs} * (\phi_{max,p} - \phi_{min,p}) \tag{12}$$

The quantity $\phi_p$ is constant across years. Also, the resulting expenditure incurred over year $i$ is equal to:

$$e_{p_i} = r_{p_i} * \phi_p \tag{13}$$

The greater the value of $Ir_p^{pp,fs}$, the lower the share of the payments attributed to expenditures incurred on that specific year. The rationale for this hypothesis is that a later reimbursement pattern most likely results from conspicuous time lags between the incurred expenditures and the reimbursed payments.

The residual of the payment (i.e., the fraction of the payment not attributed to expenditures incurred in the same year) is spread onto the previous years as per the third uncertain parameter, *Years*. To demonstrate, consider the task of attributing the residual of the 2017 payment over the three previous years. In this case, a fraction of the payment reimbursed in the year 2017 is attributed to expenditures incurred in the years 2016, 2015, and 2014 under the assumption that these are halving each preceding year: specifically, $\frac{4}{7}$ of the residual of the 2017 payment is attributed to expenditures incurred in 2016, $\frac{2}{7}$ in 2015, and $\frac{1}{7}$ in 2014. The rationale behind this assumption is that the magnitude of a payment is most likely more strongly correlated with more recent expenditures.

The assumption is also made that the total number of backward years of expenditures that the residual of the payment is attributed to is correlated over the programming period; in particular, we assume that it decreases by one year each preceding year. In this way, if the residual of the 2017 payment is attributed to expenditures incurred over the three previous years, this quantity would only amount to two for the residual of the payment reimbursed in the year 2016, and one for 2015. The minimum number of backward years is one, and this quantity is kept constant for all the years backwards up to the second year of the programming period once this threshold is met. Payments reimbursed in the first year of the programming period are entirely attributed to expenditures incurred in the same year.

In total, $2^{17}$ ($\sim 130,000$) Monte Carlo simulations were performed by sampling the uncertainty parameters from these distributions through quasi-random $LP_\tau$ Sobol' low-discrepancy sequences [24]. This sample size was selected to ensure the convergence of the sensitivity indices (see the convergence plots in S1-S20 Figs in S1 File. The rationale for using Monte Carlo simulations was to generate a population of expenditure distributions and to evaluate their

reliability against figures reported by the NMAs. The Python scripts are available as Jupyter Notebooks on GitHub.

The Python scripts thoroughly describe the preparation and curation of the dataset for the comparison, as well as the uncertainty analysis performed. The output variable is the cumulative distance $\mu_{IT}^{s}$ between the reported IT expenditure and the modelled expenditure previously introduced. After consulting with practitioners involved in EU regional policy programmes, the assumption was made that the last year of eligible expenditures was 2017.

To understand how the uncertainty of the input parameters reflected onto the output uncertainty moment-independent sensitivity analysis was performed using a *Matlab*® *Betaks3* subroutine. In moment-independent sensitivity analysis, the sensitivity measure is the distance between the unconditional and conditional distributions of the uncertain parameters. The $\delta$ sensitivity measure [19, 20] is calculated for the four input parameters shown in Table 1. The logic behind $\delta$ is the following: one factor is fixed to a value, and the difference between the curve obtained by fixing this factor and the standard output curve is measured. If the factor is important, fixing it will tangibly affect the output in terms of curve shape. The experiment is repeated by fixing the factor at different values over its range of variability until an average difference is obtained.

## Results and discussion

The distance against the reported MS expenditures for the modelled expenditures and payments for Italy (IT) is illustrated in Fig 1. Mismatches between the certified and uncertified data amount to approximately 10% of the former.

In Fig 1, the boxplots are always below the threshold range defined by the distance with the reported EC payments (black rectangles), with no or a (mostly) minor degree of

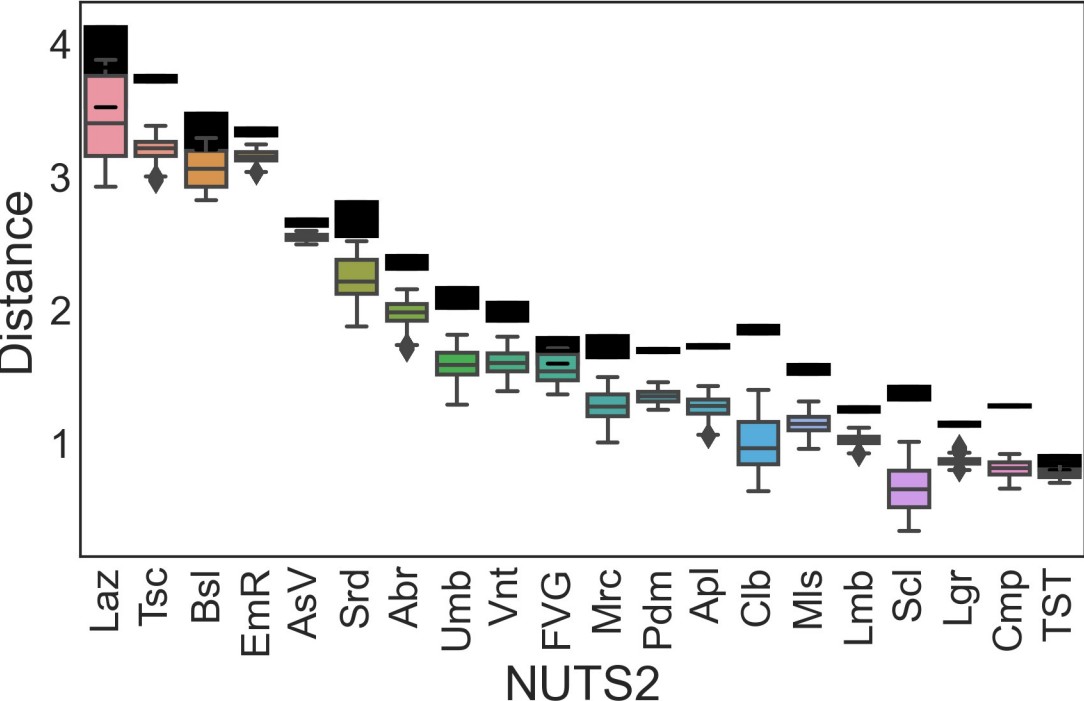

**Fig 1. Boxplots of distributions of yearly cumulative distance between modelled and reported expenditures.** Black rectangles show the range of yearly cumulative distance between the reported EC payments and reported expenditures to Italian NMAs.

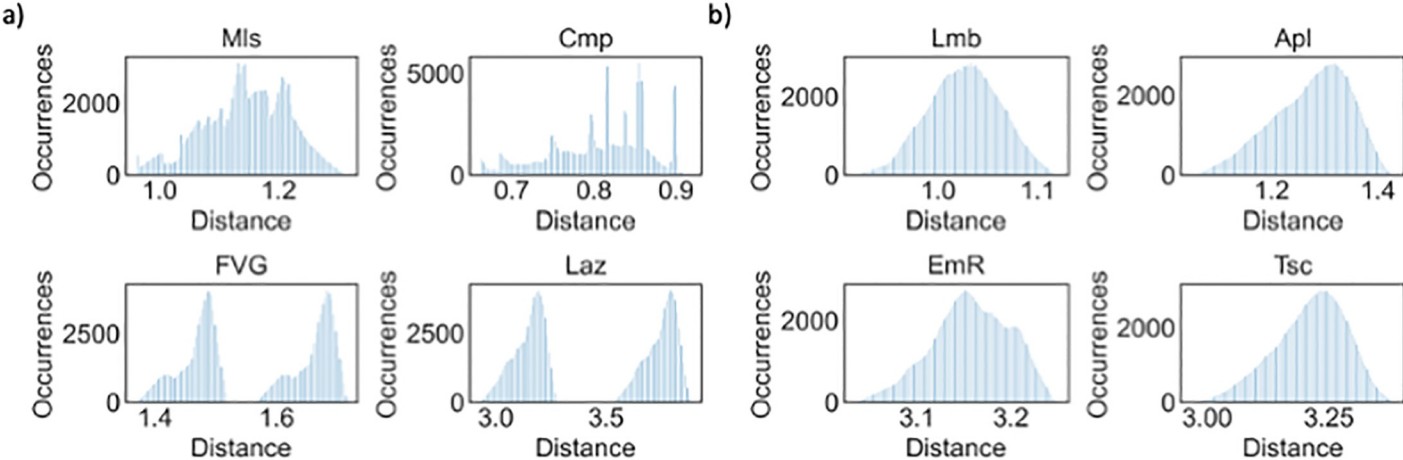

**Fig 2.** Distributions for (a) Mls, Cmp, FVG, and Laz; and (b) Lmb, Apl, EmR, and Tsc Number of occurrences in the simulation against cumulative distance between estimated and reported expenditures.

superimposition. This indicates that the model is moving the estimated expenditures in the right direction. The regions with the widest distributions are Lazio (Laz), Calabria (Clb), Sicily (Scl), and Sardinia (Srd). Conversely, Aosta Valley (AsV) and Trentino-South Tyrol (TST) are the regions with the narrowest distributions.

By examining the shapes of the distributions, it is possible to identify multi-peak and/or multi-modal distributions for certain regions. The most emblematic cases are Molise (Mls), Campania (Cmp), Friuli Venezia Giulia (FVG), and Laz, as shown in Fig 2. By contrast, Lombardy (Lmb), Apulia (Apl), Emilia-Romagna (EmR), and Tuscany (Tsc) show the most normal distributions, but they still exhibit some degree of skewness (Fig 2).

The remaining cases are provided in S21 Fig in S1 File. By examining the regional $Ir_p$, it is not possible to identify a precise correlation between this value and the distribution's width, skewness, or level of multi-modality.

Uncertainty analysis enables the investigator to apportion the output uncertainty onto the input parameters. The use of moment-independent sensitivity analysis is *a fortiori* justified by the skewed and multi-model distributions obtained, which implies that variance is a potentially poor measure of output uncertainty in these settings. The values of the sensitivity measure $\delta$ are reported in Fig 3. The higher the value, the more influential the uncertainty of the input parameter on the output uncertainty.

The variable *Residual selector* was identified as the most influential parameter for 10 of 20 Italian regions, *Years* for 8, and $\phi_{max}$ for the remainder. Figs 4 and 5 show an example for each of the two first cases, respectively. The charts for the other regions can be found in S22-S39 Figs in S1 File. In Fig 4, one can appreciate how the different values of the trigger *Residual selector* 'activate' each part of the bi-modal-shaped distribution. The trend is similar in Fig 5, where the distinct sub-components of the output distribution are activated by different values of *Years* (Fig 5). In the *y given $\phi_{min}$ sub-Figure*, one can also appreciate the importance of this parameter, given the wide range explored in the output shape upon varying this continuous parameter in its range.

This information can be used to inform a factor-prioritisation strategy [15] focused at reducing the QoI (i.e., the cumulative distance with the reported MS expenditures) by fixing the most influential parameters to the value that would minimise this quantity. At this point,

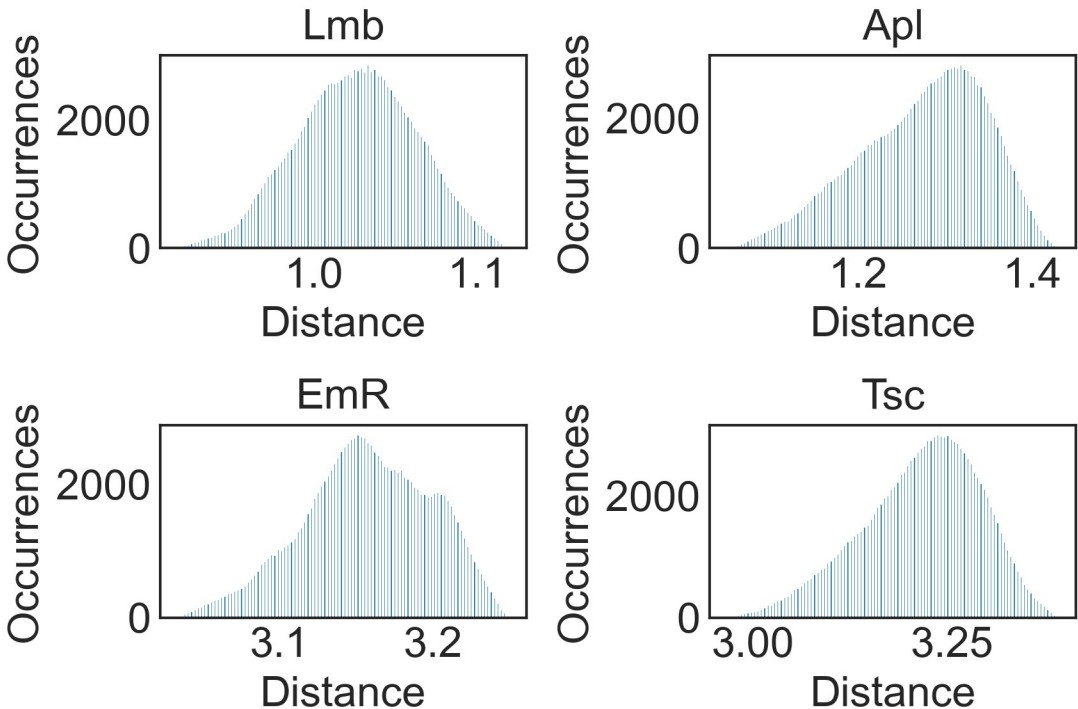

**Fig 3. Moment-independent sensitivity measure $\delta$ for uncertain parameters in the pool of Italian regions and the average figure across regions.**

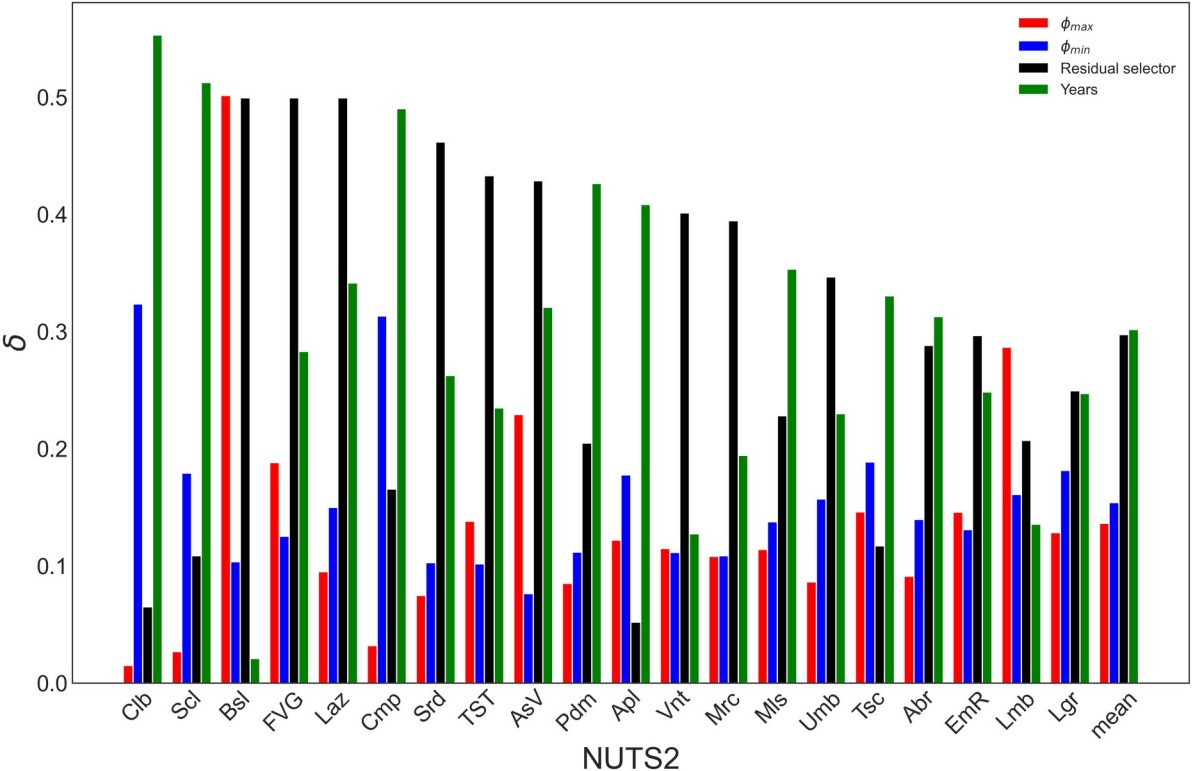

**Fig 4. Conditional output distributions for Laz when fixing the uncertain input parameters.**

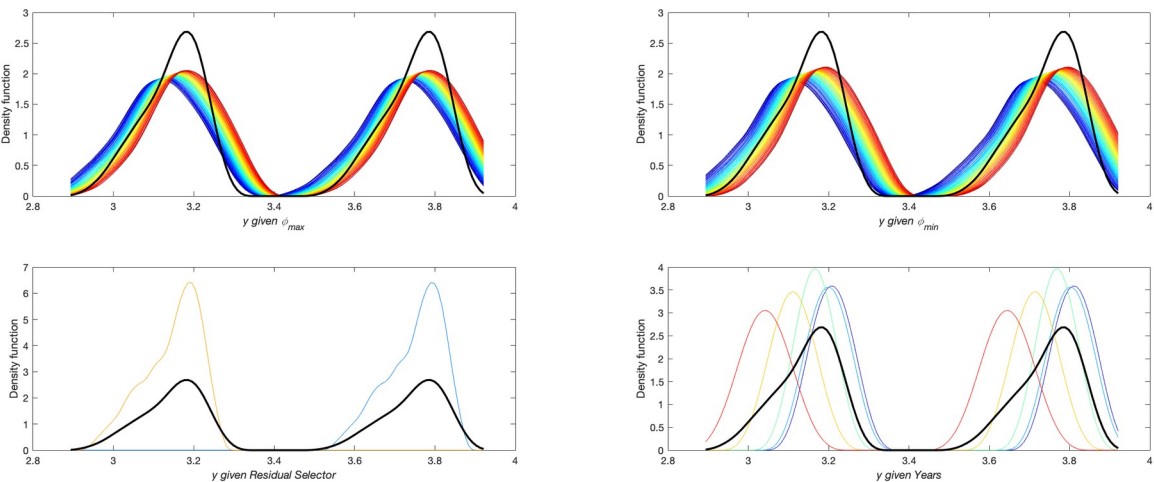

**Fig 5. Conditional output distributions for Clb when fixing the uncertain input parameters.**

let us assume that it is possible to choose either one option or the other for the trigger *Residual selector*. On doing so, the option of attributing the residual according to the reported regional yearly expenditures will lead to a lower distance in the case of FVG (Fig 6). This trend is shared by 16 of the 20 Italian regions. The trajectory is similar if the other parameter with the greatest influence, *Years*, is fixed towards its higher end, as indicated in the example of Clb in Fig 6. This trend is shared by 18 of the 20 Italian regions.

When fixing both these parameters at their optimal values, one obtains a reduction for 18 of the 20 regions. Notably, however, Laz is one of the two regions showing an opposite trend.

These findings are encouraging, yet the sample of regions investigated is too small to conclude that fixing these factors is an effective strategy to reduce the output variability and simplify the model developed: figures for more MSs would be needed to draw robust inferences that corroborates potential adjustments.

Existing interactions among the parameters justifies the choice of global sensitivity analysis over a plain one-variable-at-a-time approach, that would overlook them. Using variance-based

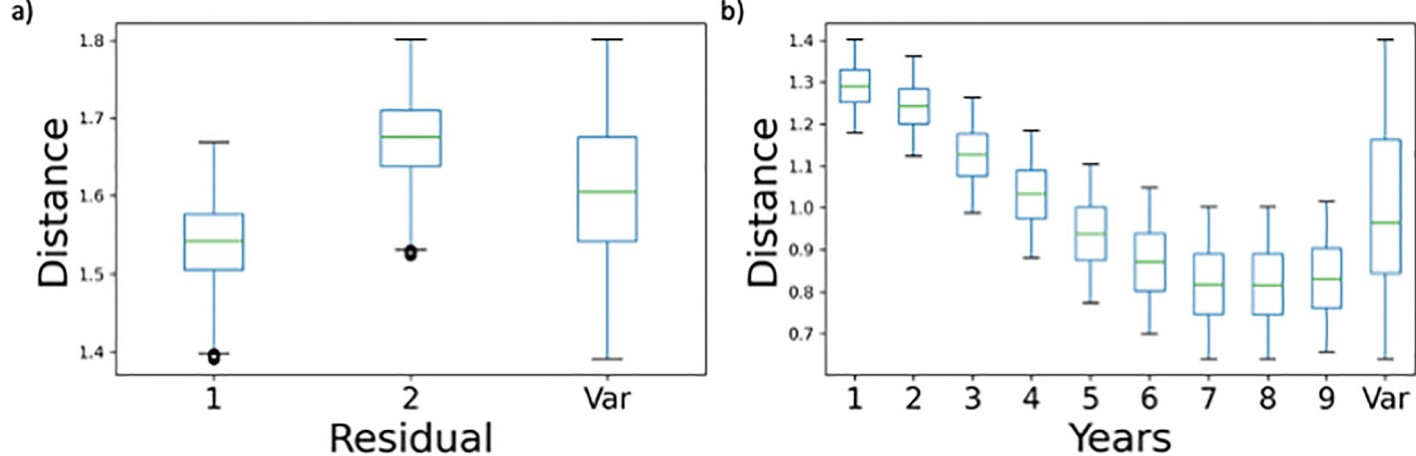

**Fig 6.** Conditional distributions of distance against the initial distribution of (a) FVG on *Residual selector*; and b) Clb on *Years*.

sensitivity analysis [15], where the sensitivity metric is the variance of the output cumulative distance, more than 10% of the output variance would not be attributed for two regions: Liguria (Lgr) and Cmp. Only 80% of the output variance is apportioned when neglecting interactions among factors for the latter, while this is only 55% for the former, for which interactions among factors are responsible for 45% of the output variance.

## Conclusions

This study presented a model to infer incurred expenditures for European regions from the reported reimbursement pattern of EC funds. The data-driven model elaborated more reliable local time-resolved figures based on the patterns reported in official centralised figures. In our study, we also validated the output time-resolved database through benchmarking against the local official statistics. We showcased an application of our model for the ERDF by considering the example of Italy for the programming period 2007–2013.

The work presented here aligns closely with several of the core principles of the European Statistics Code of Practice [25], where the common quality framework of the European Statistical System for the National Statistical Authorities and Eurostat is defined. In particular, it is consistent with Principle 4, Commitment to Quality, because it develops procedures to monitor and improve the quality of the statistical data processes, also favouring the integration of data from multiple sources. It is also aligned with Principle 12, Accuracy and Reliability, through its assessment and validation of source data, integrated data, intermediate results, and statistical outputs.

The model implemented in this study can move from EC payments to simulated expenditures in a way that matches the actual expenditures on the ground as closely as possible. Testing involved comparing the cumulative trends of the modelled and reported expenditures for the Italian regions. Perfect closure against this benchmark was not possible due to around 10% uncertified expenditures. Reducing this gap would enable analysts to assess the quality of the modelling activity performed more effectively. This modelling exercise would need to be extended across EU regions to backstop its findings, although it will never be perfect due to the existence of multiple intervening factors (e.g., payment suspensions, issues with the management of individual OPs or with the processing of the data at any level, and strategic decisions for reporting expenditures). Nonetheless, the figures produced in this study at the individual NUTS2 level for IT proved to be quite reliable.

Uncertainty and sensitivity analyses enabled this study's assessment of the range of uncertainty in the temporal discrepancies between the reported and modelled expenditures, as well as its apportionment onto the input parameters and assumptions, respectively. The most impactful assumptions turned out to be the following: first, how the non-clearly regionalised expenditures are spread onto individual regions, *Residual selector*; and second, the number of backward years onto which the residual of the yearly payments is spread, *Years*. We re-ran our simulation by keeping these variables fixed at precise points in the admissible range. The results of the sensitivity analysis for the two parameters show that both increasing the number of years of backward shift and attributing the non-clearly regionalised expenditures (on the basis of yearly regional expenditures) may improve the match between the modelled and measures expenditures. However, the gain was marginal for some regions and the fit was worse for 2 regions out of 20.

These findings are encouraging given the simplicity of the model developed. Further sophistication may be adopted to better reproduce the MS expenditures reported, as well as to corroborate this factor-prioritisation strategy.

The following are the principal take-home messages for different stakeholders:

- The EC may expand this research to other funding schemes and programming periods, including the current recovery funds for member states.

- Further MSs can make their figures available to strengthen the findings presented here. Standardised tools and forms for registering all payments would be helpful, along with data access for the researcher, ideally through central storage at the DG REGIO.

- The model's validation against MSs figures provides a new database that econometricians can use to generate less time-lagged and more reliable estimates of the benefits of European funds on the ground.

- The community of official statisticians may seek to replicate this case study in other relevant examples, including other funding schemes, international aid funds, and trade figures.

## Supporting information

**S1 File. The Figures are found in the Supporting Information file.**
(PDF)

## Acknowledgments

We would like to thank Carlo Amati, Rosaria Chifari, Roger Strand, and Roman Römisch for their contributions.

## Author Contributions

**Conceptualization:** Samuele Lo Piano, Andrea Saltelli.

**Data curation:** Samuele Lo Piano.

**Formal analysis:** Samuele Lo Piano, Emanuele Borgonovo.

**Funding acquisition:** Samuele Lo Piano, Arnald Puy, Andrea Saltelli.

**Investigation:** Samuele Lo Piano, Emanuele Borgonovo.

**Methodology:** Samuele Lo Piano, Emanuele Borgonovo, Andrea Saltelli.

**Project administration:** Samuele Lo Piano.

**Resources:** Samuele Lo Piano, John Walsh, Daniele Vidoni.

**Software:** Samuele Lo Piano, Emanuele Borgonovo.

**Supervision:** Andrea Saltelli.

**Validation:** Samuele Lo Piano, Emanuele Borgonovo, John Walsh, Daniele Vidoni.

**Visualization:** Samuele Lo Piano, Emanuele Borgonovo.

**Writing – original draft:** Samuele Lo Piano, Andrea Saltelli, Daniele Vidoni.

**Writing – review & editing:** Samuele Lo Piano, Emanuele Borgonovo, Arnald Puy, Andrea Saltelli, John Walsh.

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
