## [Decision Letter · Decision Letter 0]

16 Feb 2022

PONE-D-22-00671Improving the reliability of cohesion policy databasesPLOS ONE

Dear Prof. Lo Piano,

Thank you for submitting your manuscript to PLOS ONE. After careful consideration, we feel that it has merit but does not fully meet PLOS ONE’s publication criteria as it currently stands. Therefore, we invite you to submit a revised version of the manuscript that addresses the points raised during the review process, which are enclosed in this e-mail.

We look forward to receiving your revised manuscript.

Kind regards,

Roberto Savona

Academic Editor

PLOS ONE

Journal Requirements:

[We would like to thank Carlo Amati, Rosaria Chifari, Roger Strand, and Roman R¨omisch for their contributions. Arnald Puy also worked on this paper on a Marie Sklodowska-Curie Global Fellowship, grant number 792178.] 

 [AP worked on this paper on a Marie Sklodowska-Curie Global Fellowship (Horizon 2020 https://ec.europa.eu/programmes/horizon2020/), grant number 792178.

he funders had no role in study design, data collection and analysis, decision to publish, or preparation of the manuscript]

Reviewers' comments:

Reviewer's Responses to Questions

**Comments to the Author**

1. Is the manuscript technically sound, and do the data support the conclusions?

Reviewer #1: Yes

2. Has the statistical analysis been performed appropriately and rigorously? 

Reviewer #1: Yes

3. Have the authors made all data underlying the findings in their manuscript fully available?

Reviewer #1: Yes

4. Is the manuscript presented in an intelligible fashion and written in standard English?

Reviewer #1: Yes

5. Review Comments to the Author

Reviewer #1: The manuscript presents an innovative data-driven model to reconstruct a reliable temporal pattern for time-lagged statistical monetary, applied to ERDF in Italy. I found the study interesting and well written. In addition, the methodology is not only interesting itself, but it presents a novel easy-to-use tool, freely available online.

I have only some minor concerns.

Specifically, at p. 6, eq. (4), at the LHS there should be μ_p^s instead of μ_p.

European Statistics Code of Practice mentioned in the conclusions should be cited and briefly recalled for the benefit of non-expert readers.

The authors should explain what are the advantage of using their technique compared to the methodologies applied to obtain regionalized data on Cohesion funds available in the following webpage of the European Commission: https://cohesiondata.ec.europa.eu/Other/Historic-EU-payments-regionalised-and-modelled/tc55-7ysv. In particular, at this link https://ec.europa.eu/regional_policy/en/policy/evaluations/ec/2007-2013/#13 it is reported how data on Cohesion funds at regional level for period 2007-2013 were modelled, and at this link https://ec.europa.eu/regional_policy/sources/docgener/evaluation/pdf/expost2006/expenditure_final.pdf the methodology for period 2006-2006 is described.

6. PLOS authors have the option to publish the peer review history of their article (what does this mean?). If published, this will include your full peer review and any attached files.

Reviewer #1: No

---

## [Author Response · Author response to Decision Letter 0]

21 Mar 2022

The response to the reviewers is found in the document enclosed.

---

## [Decision Letter · Decision Letter 1]

29 Mar 2022

Improving the reliability of cohesion policy databases

PONE-D-22-00671R1

Dear Dr. Lo Piano,

We’re pleased to inform you that your manuscript has been judged scientifically suitable for publication and will be formally accepted for publication once it meets all outstanding technical requirements.

Kind regards,

Roberto Savona

Academic Editor

PLOS ONE

Additional Editor Comments (optional):

Reviewers' comments:

Reviewer's Responses to Questions

**Comments to the Author**

1. If the authors have adequately addressed your comments raised in a previous round of review and you feel that this manuscript is now acceptable for publication, you may indicate that here to bypass the “Comments to the Author” section, enter your conflict of interest statement in the “Confidential to Editor” section, and submit your "Accept" recommendation.

Reviewer #1: All comments have been addressed

2. Is the manuscript technically sound, and do the data support the conclusions?

Reviewer #1: Yes

3. Has the statistical analysis been performed appropriately and rigorously? 

Reviewer #1: Yes

4. Have the authors made all data underlying the findings in their manuscript fully available?

Reviewer #1: No

5. Is the manuscript presented in an intelligible fashion and written in standard English?

Reviewer #1: Yes

6. Review Comments to the Author

Reviewer #1: (No Response)

7. PLOS authors have the option to publish the peer review history of their article (what does this mean?). If published, this will include your full peer review and any attached files.

Reviewer #1: No

---

## [Editor Report · Acceptance letter]

14 Apr 2022

PONE-D-22-00671R1

Improving the reliability of cohesion policy databases

Dear Dr. Lo Piano:

I'm pleased to inform you that your manuscript has been deemed suitable for publication in PLOS ONE. Congratulations! Your manuscript is now with our production department.

Kind regards,

on behalf of

Prof. Roberto Savona

Academic Editor

PLOS ONE